# GRAPH TRANSFORMER

## ABSTRACT

Graph neural networks (GNN) have gained increasing research interests as a mean
to the challenging goal of robust and universal graph learning. Previous GNNs
have assumed single pre-fixed graph structure and permitted only local context en-
coding. This paper proposes a novel Graph Transformer (GTR) architecture that
captures long-range dependency with global attention, and enables dynamic graph
structures. In particular, GTR propagates features within the same graph structure
via an *intra-graph message passing*, and transforms dynamic semantics across
multi-domain graph-structured data (e.g. images, sequences, knowledge graphs)
for multi-modal learning via an *inter-graph message passing*. Furthermore, GTR
enables effective incorporation of any prior graph structure by weighted averaging
of the prior and learned edges, which can be crucially useful for scenarios where
prior knowledge is desired. The proposed GTR achieves new state-of-the-arts
across three benchmark tasks, including few-shot learning, medical abnormality
and disease classification, and graph classification. Experiments show that GTR is
superior in learning robust graph representations, transforming high-level seman-
tics across domains, and bridging between prior graph structure with automatic
structure learning.

## 1    INTRODUCTION

There is an increasing research interest in graph neural networks (GNN). Compared to traditional
convolutional networks that are usually limited to grid-structured data, and recurrent networks that
are designed for sequence modeling, GNN has demonstrated superiority on modeling dynamic and
complex graph-structured data. On one hand, as a mainstream category of graph neural networks,
spectral approaches such as graph convolutional neural networks (GCN) (Kipf & Welling, 2017;
Defferrard et al., 2016) generalize traditional convolutional networks in the spectral domain, and
employ local receptive fields for robust graph learning. However, these approaches are restricted to
fixed graph structure, and require costly matrix computation (Zhang et al., 2018b) such as eigen de-
composition. The significant shortcoming of GCNs, namely their lack of global context for updating
nodes state which prevents long-range relation modeling, as well as the equal importance of nodes
during aggregation, also limit its wide application. On the other hand, GNNs equipped with atten-
tion mechanism, such as graph attention networks (GAT) (Velickovic et al., 2018), have emerged as
an effective alternative to GCNs by alleviating these issues. GAT incorporates self-attention mech-
anism (Vaswani et al., 2017) which selectively attends to connected graph nodes according to their
importance to the considered node. Despite these advantages, GATs, the same as many existing
graph neural network approaches (Niepert et al., 2016; Zhang et al., 2018b), are still limited to
evolution within the same graph structure, and lack of the capability of transforming representa-
tions across different graph structures, or various data domains (e.g. images, sequences, knowledge
graphs, and 3D meshes).

The recently proposed Transformer (Vaswani et al., 2017) architecture for sequence modeling has
been proven useful for tasks such as machine translation (Bahdanau et al., 2017). By addressing
the inherent sequential computation shortcoming of recurrent neural networks, Transformer enables
efficient and paralleled computation by invoking a self-attention mechanism for global context mod-
eling. Transformer also avoids vanishing gradients problem (Hochreiter et al., 2001), and automat-
ically approximates relations among a dynamic range of symbols, which is crucial for learning
real-world data of arbitrary structures.

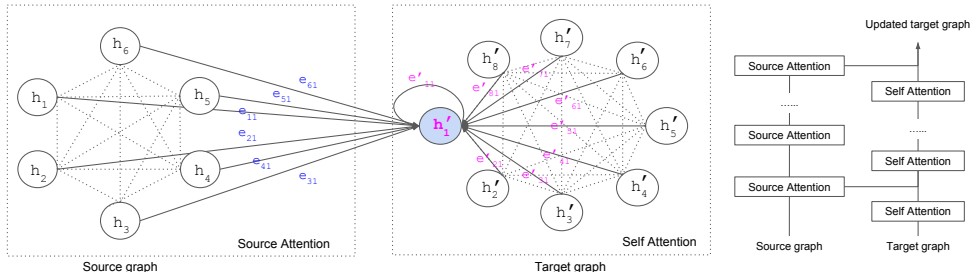

Figure 1: Illustration of *Graph Transformer* on updating a target graph node (left panel) and its stacked version (right panel). $h_i$ and $h_i'$ indicates latent state of the $i_{th}$ source graph node and the $i_{th}$ target graph node respectively, where source graph and target graph are two different graphs. $e_{ij}$ indicates edge connecting source node $i$ and target node $j$. GTR evolves a target graph by recurrently performing *Source Attention* on a source graph in an *inter-graph message passing* process, and *Self Attention* on target graph in an *intra-graph message passing* process.

Inspired by this work, we proposed a novel Graph Transformer (GTR) architecture which is a straightforward yet effective extension of the sequential Transformer architecture to a variety of complex graph-structured data, such as knowledge graphs, images, and sequences. The goal of GTR is to transform a source graph to a target graph of possibly different structures by considering global context and realizing computation parallelism across all graph nodes. An illustration is provided in Figure 1. GTR invokes an *intra-graph message passing* paradigm to progressively refine graph representations within the same graph structure, and an *inter-graph message passing* paradigm to attentively distill related semantics from source graphs for target graph learning. By alternatively propagating information either within the same graph structure or from the source graph to the target graph, GTR is able to extract key information embedded in the source graph, and refine target graph representations by considering both local and global context.

Our contributions are three-fold: 1) We propose a novel Graph Transformer (GTR) which not only learns graph representations of the same structure, but also across different structures. 2) We propose formulations of converting various real-world data types (e.g. images, sequences, knowledge graphs) to graph-structured, and the application of GTR on converting semantics across these various types of data. 3) We propose an effective way of incorporating prior graph structure with learning-based structure, and thus reconciles traditional knowledge-based and modern learning-based approaches in the down-stream tasks (e.g. few-shot learning, medical abnormality and disease classification).

We conduct experiments on three benchmark tasks: few-shot learning, medical abnormality and disease classification, and graph classification. Experiments show that GTR achieves new state-of-the-arts on all tasks, demonstrating its superior capability in learning robust graph representations, propagating information across multi-domain data types, and effective utilization of prior knowledge. Specifically, on 1-shot learning on miniImageNet dataset, GTR improves the state-of-the-art by 1.32% by incorporating category relations in novel class weight generation. GTR also improves abnormality classification accuracy by more than 7% on all evaluated data sets, demonstrating its effectiveness on knowledge graph representation learning and transformation. And, on graph classification task, GTR outperforms all compared standard graph neural networks and kernel-based networks with roughly the same computational cost, demonstrating its efficiency.

## 2    RELATED WORK

Graph neural networks (GNN) have attracted increasing research interests (Defferrard et al., 2016; Kipf & Welling, 2017; Monti et al., 2017) and provided flexible representation learning of real-world graph-structured data. There has been work that study message passing in graph learning (Do et al., 2018; Gilmer et al., 2017; Schlichtkrull et al., 2018), and equip GNNs with attention mechanisms as which is shown powerful for sequence and image modeling (Zhang et al., 2018a; Vaswani et al., 2017; Al-Sabahi et al., 2018). However, most existing methods learn to encode the input feature into higher-level feature through selective attention over the object itself (Wang et al., 2018a; Velickovic

et al., 2018), while our method works on multiple graphs, and models not only the data structure within the same graph but also the transformation rules among different graphs.

Among all related works, Transformer (Vaswani et al., 2017) can be formulated as a particular instance of Graph Transformer where input and output are both sequences, and no prior graph structure is provided. Besides, GTR is close to graph attention networks (GAT) (Velickovic et al., 2018) in that they both employ attention mechanism for learning importance-differentiated relations among graph nodes. However, GTR differs from GAT in several aspects. First, GTR attends to all graph nodes at every graph update disregard of whether two nodes are directly connected or not, while GAT only attends to directed connected neighboring nodes for a considered node by a proposed *masked attention*. Thus, GAT is still limited to differentiating importance among locally connected nodes. GTR, on the other hand, is able to capture global context as which is significant for modeling long-range relations and fasten graph learning by allowing information propagation between implicitly connected nodes. Second, GTR incorporates both prior graph structure and graph structure learning where using prior edge weights allows the efficient utilization of prior knowledge. Third, GTR is able to transform representations across various graph structures while GAT is restricted to the same graph propagation.

## 3 GRAPH TRANSFORMER

Graph Transformer (GTR) transforms a source graph to a target graph by encoding features of the same graph structure into higher-level semantics, and fusing latent semantics between different graphs (e.g. from visual features to knowledge graphs). We represent a graph as $G = (V, E)$ where $V = \{\mathbf{v}_i\}_{i=1:N}$ is a set of nodes with each $\mathbf{v}_i \in \mathbb{R}^d$ representing a node's feature of dimension $d$, and $N$ is the number of nodes in the graph. $E = \{e_{i,j}\}_{i,j=[1,N]}$ represents a set of edges between any possible pair of nodes. Here we consider the setting where each edge is associated with a scalar value indicating closeness between nodes, while it is straightforward to extend the formalism to other cases where edges are represented as non-scalar values such as vectors.

GTR takes a source graph $G = (V, E)$ as input, and transforms it to a target graph $G' = (V', E')$, where $G$ and $G'$ are two different graphs and can have different structures and characteristics (e.g. $N \neq N'$, $d \neq d'$, and $e_{i,j} \neq e'_{i,j}$). This property differs from many previous methods (Defferrard et al., 2016; Kipf & Welling, 2017; Velickovic et al., 2018) which are restricted to the same graph structures. For both source and target graph, the set of nodes $V$ and $V'$ has to be given in prior, such as the vocabulary size when the considered graph is sequences, and abnormality nodes if the considered graph is an abnormality graph. We consider two scenarios for the edges among graph nodes: 1) the edges are given in prior, and denoted as $e_{s_i, t_j}$ where $s_i$ is the $i_{th}$ node of source graph and $t_j$ is the $j_{th}$ node of target graph; 2) the edges are unknown, and thus source and target nodes are represented as fully connected with uniform weights. We assume $e_{s_i, t_j}$ as normalized, to avoid notation of averaging. Two types of message passing are considered in GTR: the one from source graph to target graph denoted as *inter-graph message passing*, and another within the same graph denoted as *intra-graph message passing*.

### 3.1 INTER-GRAPH MESSAGE PASSING

To distill relevant information from a source graph, the features of source nodes are transformed and passed to target nodes with their corresponding edge weights, which can be formulated as:

$$\mathbf{v}'_j = \mathbf{v}'_j + \sigma\left(\sum\nolimits_{i=1}^{N} e_{s_i, t_j} \mathbf{W}_s \mathbf{v}_i\right) \tag{1}$$

where $\sigma$ is a nonlinear activation, and $\mathbf{W}_s$ is a projection matrix of size $d' \times d$.

Considering that the edge information between source and target graphs may not be available in many cases such as translating a sequence of words into another sequence of words, we propose to learn edge weights automatically by an attention mechanism (Vaswani et al., 2017). By doing so, target node update is enabled to consider the varying importance of the source nodes in global context. Specifically,

$$\hat{e}_{s_i, t_j} = Attention(\mathbf{W}_s^a \mathbf{v}_i, \mathbf{W}_t^a \mathbf{v}'_j) \tag{2}$$

where $\hat{e}_{s_i, t_j}$ is the attention weight of edge from source node $i$ to target node $j$; $\mathbf{W}_s^a$ and $\mathbf{W}_t^a$ are weights in attention mechanism to project nodes features of source graph and target graph to a

common space of dimension $q$ respectively; and *Attention*: $\mathbb{R}^q \rightarrow \mathbb{R}$ is the attention mechanism that transforms the two projected features $\mathbf{W}_s^a \mathbf{v}_i$, $\mathbf{W}_t^a \mathbf{v}_j' \in R^q$ to a scalar $\hat{e}_{s_i,t_j}$ as the edge's attention weight. In our experiments, *Attention* is parameterized as a scaled dot-product operation with multi-head attention (Vaswani et al., 2017).

The attention weights are normalized across all source nodes for each target node, representing the relative importance of each source node to a target node among all source nodes. The formulation can be written as:

$$\hat{e}_{s_i,t_j} = softmax_{s_i}(\hat{e}_{s_i,t_j}) = \frac{\exp\left(\hat{e}_{s_i,t_j}\right)}{\sum_{k=1}^{N} \exp\left(\hat{e}_{s_k,t_j}\right)} \tag{3}$$

Once obtained, the normalized attention coefficients are be combined with prior edge weights in order to pass features of connected source nodes to target nodes. The combined features serve as the target node's updated features with source graph knowledge encoded. We adopt weighted sum of the learned attention edge weights and prior edge weights as final edge weights. Other methods such as multiplication of the learned and prior edge weights followed by softmax also works. However, in our experiments, we observed that the first method performs better and avoids under-fitting. The formulation can be written as:

$$\tilde{e}_{s_i,t_j} = \lambda e_{s_i,t_j} + (1 - \lambda)\hat{e}_{s_i,t_j} \tag{4}$$

where $\lambda$ is a user-defined weight controlling importance of prior edges and learned edges. If $\lambda$ is set to 1, the edges between source graph and target graph are fixed, and no attention machanism is required. The formulation is then the same as Equation 1. If $\lambda$ is set to 0, the edges between source graph and target graph are completely learned by the model. With the updated weight, one can obtain updated target nodes features via Equation 1.

### 3.2 Intra-graph message passing

Intra-graph message passing aims at modeling the correlation among nodes of the same graph, and fusing features according to the closeness between them. Specifically, a target node is updated by combining features of neighboring nodes and itself. The formulation can be written as:

$$\mathbf{v}_j' = \mathbf{v}_j' + \sigma\left(\sum_{i=1}^{N'} \tilde{e}_{i,j} \mathbf{W}_t \mathbf{v}_i'\right) \tag{5}$$

where $\mathbf{W}_t$ is weight to project features of target nodes from dimension $d$ to output dimension. To learn the edge weights through attention mechanism, one can directly apply Equations 1-4 by changing source and target nodes notation to be of the same graph.

### 3.3 GTR as a module/building block

As shown in Figure 1, we formulate GTR as a module denoted as *GTR* by first concatenating intra-graph message passing and inter-graph message passing into one step (that is, first conduct message passing within the target graph, then conducting message passing from a source graph), then stacking multiple such steps into one module in order to progressively convert target graph features into high-level semantics. To denote different variants of *GTR* for different input or output domains, we use $s$ to denote sequence, $i$ to denote images, and $g$ to denote knowledge graphs, and represent *GTR* for image input and knowledge graph output as $GTR_{i2g}$, *GTR* for knowledge graph input and knowledge graph output as $GTR_{g2g}$ and the rest possible combinations in the same manner.

### 3.4 GTR for Multiple Domains

Most real-world data types (e.g. images, sequences, graphs) can be formulated as graph-structured. For example, a 2-dimensional image can be formulated as a graph whose nodes are pixels of the image where every node is connected with its neighboring pixel nodes; and a sequence of words can be formulated as a graph whose nodes are the individual words where edges among nodes are the consecutive relation among words. If the global context of the data is considered, which is commonly adopted in attention mechanism (Vaswani et al., 2017), the graph nodes are then fully-connected. In the following, we describe the variants of *GTR* for different data domains by first formulating data as graph-structured, and then perform GTR operations on it.

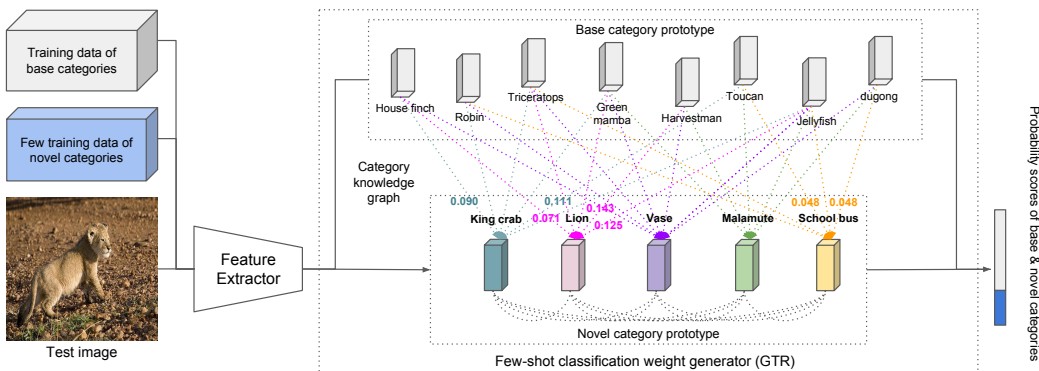

Figure 2: Architecture of *Graph Transformer* for few-shot learning. Images are first fed into a *Feature Extractor* for extracting features. The few-shot classification weight generator implemented as GTR takes a target graph whose nodes are the novel categories and features are initialized as the extracted visual features of the few training samples of novel categories, and a source graph whose nodes are the base categories and features are the corresponding category prototypes. GTR updates the target graph features by considering similarity among node features of the target graph, and that of the source graph and target graph. The final node features of the target graph are used as the generated novel category prototypes for subsequent classification.

**GTR for sequential input/output.** To apply GTR for sequential input or output (e.g. a sequence of words, a sequence of retrieved items), we employ *positional encoding* (Vaswani et al., 2017) to GTR so as to add relative and absolute position information to the input or output sequence. Specifically, we use sine and cosine functions of different frequencies:

$$PE_{pos,2i} = sin(pos/10000^{2i/d}) \tag{6}$$

$$PE_{pos,2i+1} = cos(pos/10000^{2i/d}) \tag{7}$$

where $pos$ is the position and $i$ is the dimension. If both input and output are sequences, GTR is close to a *Transformer* (Vaswani et al., 2017), however, with additional prior edge weights.

**GTR for image input.** We denote visual features of an image as $I \in R^{D,W,H}$ where D is the dimension of latent features, W and H is width and height. To apply GTR for image input, we first reshape visual features by flattening the 2-dimension into 1-dimension $R^{W \times H,D}$. Then each pixel is treated as graph node whose features are used as graph node features.

## 4 EXPERIMENTS

### 4.1 FEW-SHOT LEARNING

We evaluate GTR for few-shot learning on miniImageNet dataset (Vinyals et al., 2016) which consists of 100 ImageNet classes with each with 600 samples. We follow the same category split proposed by Ravi et al (Ravi & Larochelle, 2017) with 64, 16, 20 for training, validation and testing respectively. An N-way K-shot learning is defined as classifying N categories with each only giving K training examples. We conduct 5 way 1-shot and 5-shot classification and develop our model based on the framework proposed by Gidaris & Komodakis (2018). First, the few-shot learning comprises two stages: base category classification, and episodic learning. During base category classification learning, the images of the base categories are sampled and used for training a set of prototypes with each corresponding to a base category. The forward procedure includes feature extraction via a feature extractor, and a classification operation such as dot product and cosine similarity to classify images. During episodic learning, specifically testing, N categories are first sampled from novel categories, each with K images as examples. The K images are first fed to feature extractor to extract features, then used as initialization of novel class prototypes. The initialized novel class prototypes are further used by an attention block proposed by Gidaris & Komodakis (2018) to selectively attends to base class prototypes in order to learn novel prototypes without forgetting base category prototypes.

Gidaris & Komodakis (2018) updates novel category prototypes by selectively attending to available base category prototypes, and modeling the data-dependent relations among the base and novel

| Model | Accuracy | |
|---|---|---|
| | 1-shot | 5-shot |
| Matching networks (Vinyals et al., 2016) | 43.56 ±0.84 % | 55.31 ±0.73 % |
| Prototypical networks (Snell et al., 2017) | 49.42 ±0.78 % | 68.20 ±0.66 % |
| Meta-learner LSTM (Ravi & Larochelle, 2017) | 43.44 ±0.77 % | 60.60 ±0.71 % |
| MAML (Finn et al., 2017) | 48.70 ±1.84 % | 63.11 ±0.92 % |
| LLAMA (Grant et al., 2018) | 49.40 ±1.83 % | - |
| REPTILE (Nichol & Schulman, 2018) | 49.97 ±0.32 % | 65.99 ±0.58 % |
| PLATIPUS (Finn et al., 2018) | 50.13 ±1.86 % | - |
| SNAIL (Mishra et al., 2018) | 55.71 ±0.99 % | 68.88 ±0.92 % |
| Gidaris & Komodakis (2018) | 56.32 ±0.86 % | 73.00 ±0.64 % |
| Bauer et al. (2018) | 56.30 ±0.40 % | 73.90 ±0.30 % |
| Meta network (Munkhdalai & Yu, 2017) | 57.10 ±0.70 % | 70.04 ±0.63 % |
| TADAM (Oreshkin et al., 2018) | 58.50 ±0.30 % | **76.70 ±0.30** % |
| Qiao et al. (2017) | 59.60 ±0.41 % | 73.74 ±0.30 % |
| LEO (Rusu et al., 2018) | 60.06 ±0.05 % | 75.72 ±0.18 % |
| GTR | **61.58 ±0.69** % | 73.21 ±0.63 % |

Table 1: Accuracy on miniImageNet for 5-way 1-shot and 5-shot classification. For GTR episodic training, we use initial learning rate 0.01, and decrease by 10 times when encountering validation performance plateau. Feature extractor is fixed during episodic training. The weight on prior edges $\lambda$ (in Equation 4) is chosen by validation (see Table 6).

categories in each episodic learning. However, machine learning models are often fooled by visual illusions. For example, images of different animals with the same color and similar posture taken in the same environment may have higher probability of being classified to the same class than images of the same animal postured differently and located in the different environment. When training samples are few and visual features may not be a reliable source, having the prior knowledge of how similar categories are can help improve visual recognition drastically. To achieve this, we propose to replace the novel category weight generation process in Gidaris & Komodakis (2018) with our GTR incorporated with a prior category relation graph. The architecture is shown in Figure 2. During an episodic learning, we treat the base and novel categories as graph nodes, and replace the attention mechanism in the weight generator in Gidaris & Komodakis (2018) with GTR. We define the input source graph of GTR as the base categories whose features are their prototypes, and the target graph as novel categories whose features are initialized by the visual features of the K examples as initial target graph. GTR evolves the target graph by sequentially attending to source graph and itself. We further define the edges among different graph nodes as the similarity between the category words computed by $path\_similarity()$ API of wordnet (Miller, 1995) which returns the shortest path between two nodes in the is-a taxonomy. To encourage sparsity, we further prune edges with smaller than 0.1 scores, and normalize the remaining out-going edges per node. The similarity scores are used as prior edge weights, and are weighted averaged with the learned edges weights during training and inference. We denote the importance weight of prior edges as $\lambda$ (see Equation 4), and experiment different values for $\lambda \in \{0.3, 0.4, 0.5, 0.6\}$. The GTR in our experiment has 3 layers and 6 heads, and is trained with 0.1 dropout, and maximum 60 epochs. For base category classification, we use 0.1 initial learning rate, and decrease it to 0.06, 0.012, 0.0024 at epoch 20, 40, 50 respectively. For episodic training, we use 0.01 learning rate. We use ConvNet128 as proposed in Gidaris & Komodakis (2018) for extracting visual features of dimension $128 \times 5 \times 5$.

**Results.** We compare to a range of state-of-the-art methods including that use convolutional networks and deep residual networks as shown in Table 1 along with the results. GTR achieves a new state-of-the-art result on 1-shot learning, and has improved on 5-shot learning compared to Gidaris & Komodakis (2018). Furthermore, GTR obtains larger improvements on 1-shot than 5-shot, demonstrating that the incorporation of prior category relations takes larger effect when less training samples are given, and thus prior knowledge from textual semantic domains helps the model learn novel visual patterns more effectively and accurately.

## 4.2 MEDICAL ABNORMALITY & DISEASE CLASSIFICATION

To demonstrate the capability of GTR on transforming graph features between graph-structured data of different domains such as from image features to knowledge graph, and from knowledge graph to another knowledge graph, we conduct the experiment on medical abnormality and disease clas-

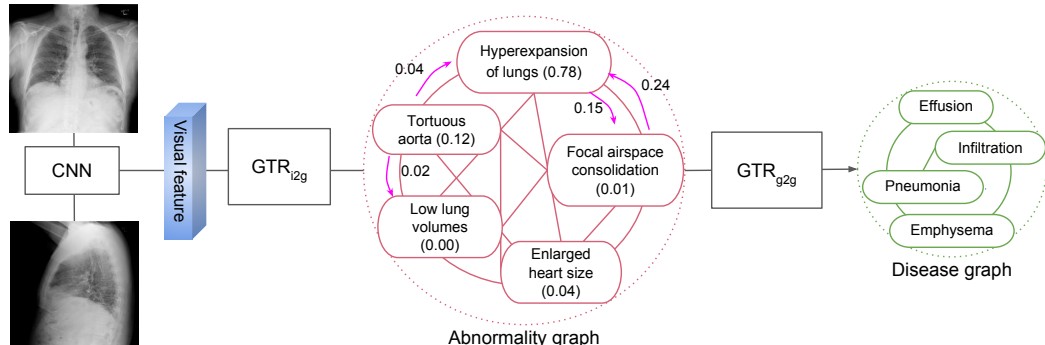

Figure 3: Architecture of *Graph Transformer* for medical abnormality and disease classification. A $GTR_{i2g}$ is first used to convert visual features extracted by CNN from images to an abnormality graph whose nodes are potential medical abnormalities. A $GTRg2g$ then transforms the abnormality graph to a disease graph by considering correlations among diseases and abnormalities.

sification. Specifically, we formulate the abnormality and disease categories as graph nodes where the edges among nodes represent their correlation (e.g. co-occurrence frequency). The learning of abnormality and disease classification is thus learning abnormality and disease graph features whose semantics represent clinical conditions that could lead to the diagnosis of abnormalities or diseases. In detail, we first compile an off-the-shelf abnormality graph that contains frequent abnormalities stem from thoracic organs. For example, "disappearance of costophrenic angle", "low lung volumes", and "blunted costophrenic angle". Additionally, we design a disease graph containing common *thorax* diseases (e.g. nodule, pneumonia and emphysema) which are commonly concluded from the single or combined condition of abnormalities. For example, atelectasis may be concluded if "interval development of bandlike opacity in the left lung base" is present;

As described in Figure 3, a set of images are first fed into a CNN for extracting visual features which are then transformed into an abnormality graph via $GTR_{i2g}$. A $GTR_{g2g}$ module then converts the abnormality graph into a disease graph. The node features of abnormality graph and disease graph are passed to separate classification layers for predicting abnormality and disease categories respectively.

We conduct experiments on two medical image datasets. First, **Indiana University Chest X-Ray Collection (IU X-Ray)** (Demner-Fushman et al., 2015) is a public dataset consisting of 7,470 chest x-ray images paired with their corresponding diagnostic reports. Each patient has a frontal view and a lateral view image, a report, and corresponding MeSH (Frey & George, 2007) tags. We text-mined 80 most frequent abnormality labels and 14 disease labels from reports and MeSH tags respectively. Examples of abnormality and disease labels are shown in Figure 3 and Table 3. **CX-CHR** is a private dataset of chest X-ray images collected from a professional medical institution. The dataset consists of 35,609 patients and 45,598 images. Each patient has one or multiple chest x-ray images in different views (e.g. frontal and lateral), and a corresponding Chinese report. We select patients with no more than 2 images and obtain 33,236 patient samples in total which covers over 93% of the dataset. We text-mine most popular abnormalities and diseases mentioned in reports, which yields 155 abnormalities and 14 most popular thorax diseases. On both datasets, we randomly split the data by patients into training, validation and testing by a ratio of 7:1:2. There is no overlap between patients in different sets.

For abnormality classification, we compare our GTR with a DenseNet (Huang et al., 2017) on classifying the same set of abnormality labels. For ablation study, we compare our method by solely training on abnormality classification (GTR-1graph), and jointly training on both abnormality and disease classification (GTR-2graphs). For disease classification, we compare our method with DenseNet on CX-CHR dataset, and TieNet (Wang et al., 2018b) on IU X-Ray dataset. We use learning rate $1e^{-3}$ for training and $1e^{-5}$ for fine-tuning, and reduce by 10 times when encountering validation performance plateau.

**Results.** The area under the curve (AUC) of abnormality classification is shown in Table 2, and AUC of disease classification is shown in Table 3. On both abnormality and disease classification, GTR achieves the best results on both dataset, demonstrating its superior capability of learning and transforming graph features.

| IU X-Ray | | | CX-CHR | | |
|---|---|---|---|---|---|
| DenseNet | GTR-1graph | GTR-2graphs | DenseNet | GTR-1graph | GTR-2graphs |
| 0.612 | 0.674 | **0.686** | 0.689 | 0.721 | **0.760** |

Table 2: Averaged AUC of abnormality classification.

| IU X-Ray | | CX-CHR | |
|---|---|---|---|
| TieNet (Wang et al., 2018b) | GTR-2graphs | DenseNet (Huang et al., 2017) | GTR-2graphs |
| 0.719 | **0.727** | 0.800 | **0.862** |

Table 3: Averaged AUC of disease classification.

| Model Category | Model | PROTEINS | D&D |
|---|---|---|---|
| Kernel-based | AWE (Ivanov & Burnaev, 2018) | - | 71.51 |
| | FGSD (Verma & Zhang, 2017) | 73.42 | 77.10 |
| | PK (Neumann et al., 2012) | 73.68 | 78.25 |
| | GRAPHLET (Shervashidze et al., 2009) | 72.91 | 74.85 |
| | WL (Shervashidze et al., 2011) | 73.76 | 74.02 |
| | WL-OA (Kriege et al., 2016) | 75.26 | 79.04 |
| GNN-based | DCNN (Atwood & Towsley, 2016) | 61.29 | 58.09 |
| | DGK (Yanardag & Vishwanathan, 2015) | 71.68 | - |
| | PATCHYSAN (Niepert et al., 2016) | 75.00 | 76.27 |
| | GRAPHSAGE (Hamilton et al., 2017) | 70.48 | 75.42 |
| | ECC (Simonovsky & Komodakis, 2017) | 72.65 | 74.10 |
| | SET2SET (Vinyals et al., 2015) | 74.29 | 78.12 |
| | GTR | **75.70** | **79.15** |

Table 4: Graph classification accuracy in percent.

## 4.3 GRAPH CLASSIFICATION

To probe the ability of Graph Transformer on transforming same graph features from low-level semantics to high-level semantics, we conduct the experiment on two benchmark protein datasets for graph classification: PROTEINS (Borgwardt et al., 2005; Feragen et al., 2013) and D&D (Dobson & Doig, 2003). PROTEINS contains 1113 graphs and 43472 nodes in total. Each graph has nodes representing secondary structure elements (SSEs), and edges representing neighboring relations in the amino acid sequence or in 3D space (Borgwardt et al., 2005). The maximum number of nodes in a graph is 620, the average number of nodes is 39.06, and the average number of edges is 72.8. D&D contains 1178 protein graphs and 334926 nodes in total. The graph nodes representing amino acids are connected if they are less than 6 Angstroms apart (Dobson & Doig, 2003). The maximum number of nodes in a graph is 5748, the average number of nodes is 284.32.

We use 1e-4 learning rate and train maximum 20 epochs on one GPU. The graph nodes of both data sets are first re-organized as its unique BFS ordering before feeding into the model. We use 3 layers, 6 heads and 0.5 dropout in GTR. We pad or trim number of nodes to 100 and 500 during training and inference for PROTEINS and D&D dataset respectively. We take mean of all nodes features after GTR and feed the averaged features to a linear fully-connected layer for prediction graph labels.

**Results.** We compare with 6 kernel-based methods and 6 deep learning-based methods as shown in Table 4. We report averaged classification accuracy of 10-fold cross-validation in Table 4. Compared with kernel-based and graph neural network-based (GNN-based) baselines, our proposed method GTR achieves the best graph classification performance on both datasets, demonstrating the capability of GTR on classifying large-scale and dynamic graphs with simple architecture and avoidance of sophisticated kernels or network design.

## 5 CONCLUSION

This paper introduces the Graph Transformer that generalizes the Transformer model to multi-domain graph-structured data types, and enables the incorporation of prior knowledge. The proposed Graph Transformer achieves state-of-the-art results on three challenging benchmark tasks, namely few-shot learning, medical abnormality and disease classification, and graph classification. Graph Transformer reconciles traditional knowledge-based approaches with modern learning-based approaches, and demonstrates its significant contributions to many tasks such as few-shot learning, and abnormality and disease classification.

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

# Appendices

## A    RELATED WORK

Our work is also close to Graph Attentional Multi-Label Learning (GAML) (Do et al., 2018) which studies graph representation learning via attention mechanism. However, our work differs in that (1) GAML is proposed specifically for multi-label classification while GTR can not only be formulated for multi-label classification, but also for graph-level classification and few-shot classification which is shown in our three diverse experiments. Furthermore, GTR uses different mechanism for multi-labeling by expanding extra classification layers to each graph node while GAML treats all class labels are auxiliary nodes which are modeled in parallel with input graph nodes. (2) GAML does not actually study transformation from one graph to another but only intra-graph message passing by formulating all label nodes and input nodes into one unified graph. However, GTR formulates input nodes and label nodes as separate graphs, and studies both intra- and inter-graph message passing. This formulation is structure-agnostic and allows substantial difference between input and label graphs including graph size, edges types, and graph node feature dimension. (3) GTR uses different mechanism for multi-labeling by expanding extra classification layers to each graph node while GAML treats all class labels are auxiliary nodes which are modeled in parallel with input graph nodes. (4) GTR also can be easily extended to existing deep learning models such as DenseNet (Huang et al., 2017) and ResNet (He et al., 2016) without significant change to the deep network so as to explicitly and effectively uncover label relations and feature-label relations. On the contrary, GAML requires extra configuration of the last layer of existing deep networks in order to merge all labels nodes and features nodes into the same graph, and provides no flexibility to feature dimension or edge types to the extended GAML. Our extensive set of experiments verify that GTR serves better purposes of intra- and inter- graph message passing compared to several strong baselines.

Compared to some previous/parellel work on graph transformation such as graph2vec (Narayanan et al., 2016), graph2graph (Anonymous, 2018), graph2seq (Xu et al., 2018; Venkatakrishnan et al., 2018), AWE (Ivanov & Burnaev, 2018), and FGSD (Verma & Zhang, 2017), GTR proposes a novel and unified strategy of formulating and learning from multi-modal data types such as visual features, knowledge graphs and sequential text, where graph2vec (Narayanan et al., 2016) and graph2seq (Xu et al., 2018; Venkatakrishnan et al., 2018) can be seen as particular instances of GTR with vector or sequence as output respectively. Besides, graph2graph (Anonymous, 2018) is specifically designed for multi-label classification and its output in the conducted experiments is not actually graphs. As explained in Anonymous (2018), graph2graph does not following the conventional phrasing such as Seq2Seq in which the output is a sequence, but just representing the input and output space as two separate graphs. However, GTR indeed studies the graph to graph transformation, and follows the convention where output can be graphs such as in medical abnormality and disease classification task, as well as other data types (e.g., graph-level labels in graph classification task, and classifier parameters in few-shot learning). Thus, GTR is at the core of graph to graph transformation, and shows more flexibility towards application on a diverse set of tasks. GTR is also explainable as the label prediction probability and learned edges/relations between labels can be visualized directly. Additionally, AWE (Ivanov & Burnaev, 2018) uses random walk for graph representation learning which is limited to local context encoding, and only uses predefined edge weights, while GTR enables global context encoding via global attention mechanism and incorporation of predefined edge weights and learnable edge weights. FGSD (Verma & Zhang, 2017) uses graph spectral distances for learning graph features which requires complex computation such as eigendecomposion while our method does not require such cost. Furthermore, both AWE and FGSD use SVM as algorithm for graph classification, while our method does not use any additional classifier.

## B    MEMORY COMPLEXITY

In terms of memory overhead, GTR scales linearly with the graph size thanks to the proposed global attention mechanism. GTR does not require any additional memory compared to the standard *Transformer* (Vaswani et al., 2017) and recurrent neural networks (Ranzato et al., 2016; Donahue et al., 2015). Table 5 (second column) summarizes the memory complexity for sequence and graph out-

| Output type | Memory Complexity of Each Layer | Train Time Complexity | Test Time Complexity |
|---|---|---|---|
| Sequence | O((N+M) * d) | O(1) | O(M) |
| Graph | O((N+M) * d) | O(1) | O(1) |

Table 5: Memory and time complexity of Graph Transformer. $N$ is the number of input graph nodes, $M$ is the number of output graph nodes, $d$ is the models hidden state dimension. GTR scales linearly with graph size, and uses constant time for training and testing on graph outputs.

| Model | Accuracy | |
|---|---|---|
| | 1-shot | 5-shot |
| GTR ($\lambda = 0.0$) | 56.08 ±0.79 % | 72.12 ±0.64 % |
| GTR ($\lambda = 0.1$) | 59.82 ±0.74 % | 72.41 ±0.65 % |
| GTR ($\lambda = 0.2$) | 60.73 ±0.71 % | 72.94 ±0.63 % |
| GTR ($\lambda = 0.3$) | 61.24 ±0.71 % | 72.99 ±0.62 % |
| GTR ($\lambda = 0.4$) | 61.07 ±0.71 % | 73.07 ±0.64 % |
| GTR ($\lambda = 0.5$) | 61.43 ±0.69 % | 73.14 ±0.64 % |
| GTR ($\lambda = 0.6$) | 61.17 ±0.70 % | 72.95 ±0.62 % |
| GTR ($\lambda = 0.7$) | 61.36 ±0.71 % | **73.21 ±0.63** % |
| GTR ($\lambda = 0.8$) | 60.85 ±0.70 % | 72.82 ±0.65 % |
| GTR ($\lambda = 0.9$) | **61.58 ±0.71** % | 73.09 ±0.63 % |
| GTR ($\lambda = 1.0$) | 61.56 ±0.72 % | 72.95 ±0.64 % |

Table 6: Accuracy on miniImageNet for 5-way 1-shot and 5-shot classification using different $\lambda$ values. $GTR(\lambda = x)$ indicates $GTR$ using $x$ importance weight on prior edges, and thus $(1 - x)$ importance weight on learned edges. For GTR episodic training, we use initial learning rate 0.01, and decrease by 10 times when encountering validation performance plateau. Feature extractor is fixed during episodic training.

puts where Let N is the number of input graph nodes, M is the number of output graph nodes, d is the models hidden state dimension.

## C  TIME COMPLEXITY

Compared to many graph neural networks (Defferrard et al., 2016; Kipf & Welling, 2017; Monti et al., 2017), GTR is more parallelizable and requires significantly less time to train as every graph message propagation learns the global relations of all nodes, as opposed to only transferring features to connected neighboring nodes in the case of graph convolutional networks (GCN), or propagating previous states to subsequent states sequentially in the case of recurrent neural networks (RNN). Furthermore, in case of graph output, GTR only requires constant training and testing time as the joint probability of nodes/labels is not estimated using chain rules but predicted in parallel via global attention mechanism. Table 5 (last two columns) summarizes the training time and testing time complexity. Lastly, GTR does not suffer from vanished gradients due to the same merit of global context encoding while vanished gradients being one of the biggest challenges of most RNNs and potentially GCNs where only local context encoding is enabled. Last but not least, explicitly representing latent features as knowledge graphs may hold the key to classification performance and explainability. GTR enables intuitive visualizations for better understanding of the knowledge structure and label relations, and explanation of the model behaviors.

## D  ABLATION STUDY ON WEIGHT OF PRIOR EDGES.

The results on few-shot learning using different weight of prior edges is shown in Table 6. The highest 1-shot accuracy is achieved by $\lambda = 0.9$. The highest 5-shot accuracy is achieved by $\lambda = 0.7$. The results demonstrate that: 1) GTR with prior category similarity improves few-shot learning performance over its direct baseline framework Gidaris & Komodakis (2018) on both 1-shot and 5-shot learning, and achieves the state-of-the-art performance on 1-shot learning; 2) 1-shot learning relies more on the prior knowledge of similarity between base and novel categories than 5-shot learning; 3) GTR($\lambda$=0.0) is slightly lower than that achieved by Gidaris & Komodakis (2018), however not statistically significant, indicating that the attention mechanisms in both models have similar effectiveness in this task.

| Model | F-Dim. | H-Dim. | P-Size (M) | Memory (G) | Validation Accuracy | Test Accuracy |
|---|---|---|---|---|---|---|
| Gidaris & Komodakis (2018) | 3200 | - | 10.6638 | 0.0407 | - | 56.32±0.86% |
| GTR | 3200 | 3200 | 87.2579 | 0.3409 | 62.0833±0.4356% | **61.4333±0.6942**% |
|  | 3200 | 512 | 38.0314 | 0.1486 | 62.1060±0.4386% | 60.6067±0.7231% |
|  | 3200 | 256 | 33.3432 | 0.1303 | 61.7220±0.4415% | 60.5933±0.7206% |
|  | 2048 | 512 | 30.6823 | 0.1199 | 62.3087±0.4527% | 61.0089±0.7210% |
|  | 1024 | 512 | 15.4665 | 0.0546 | **62.2427±0.4464**% | 60.7222±0.6967% |
|  | 512 | 512 | **7.8586** | **0.0278** | 62.1507±0.4427% | 60.6778±0.7095% |

Table 7: Accuracy on miniImageNet for 5-way 1-shot classification using different feature and hidden feature dimensions. We use initial learning rate 0.01, and decrease by 10 times when encountering validation performance plateau. All other hyper-parameters are fixed. We use 0.5 prior edge weights. We compare GTR with its direct baseline Gidiaris et al. (Gidaris & Komodakis, 2018). F-Dim. indicates feature dimension. H-Dim. indicates hidden feature dimension. P-Size indicates parameter size.

# E ABLATION STUDY ON MODEL SIZE

We conduct ablation study on how performance changes with parameter size/memory usage on 1-shot learning by changing feature dimensions and hidden feature dimensions, and fixing all other hyper-parameters. Table 7 summarizes the results. It can be observed that the performance drops slightly when reducing either feature dimension or hidden feature dimension. However, all changes are not statistically significant. Furthermore, the memory usage reduces greatly from 0.3409G to 0.0278G. More importantly, GTR using 512 as feature dimension and hidden feature dimension (last row of Table 7) only consumes 0.0278G memory which is less than that used by Gidaris & Komodakis (2018) (the first row of Table 7), and improved result greatly by 4.3578%. This demonstrates that GTR is not only effective, but also memory efficient. Lastly, all results are still larger than that of all baseline models, maintaining GTRs state-of-the-art performance. A line chart of model size v.s. graph size is provided in Appendix.

