# OpenReview forum: "Graph Transformer "
_ICLR.cc/2019/Conference_

### Official Review · AnonReviewer1 · 2018-11-01
**Interesting bridge paper between knowledge-based and learning approaches with good synergy**

**Rating:** 6
**Confidence:** 3

**Review:**

I am familiar with the few-shot learning literature and this particular approach is novel as far as I know. A weight generator outputs the last layer of an otherwise pre-trained model. A combination of attention over base category weights and graph neural networks is used to parametrize the generator. Results are particularly good on 1-shot miniImageNet classification, but may not be entirely comparable with previous work. Two more interesting experiments are given and have convincingly superior results (at first glance) but I am not familiar with those domains.

I still think the questions in my previous post need answers! I am willing to improve my score if clarifications are added to the paper.

Overall, the paper makes a convincing point that hand-engineered graphs and knowledge can be effectively used with learning methods, even in challenging few-shot settings.

---

> ### Author Response · Authors · 2018-11-20
> **Response to Reviewer1**
>
> Thank you for your constructive comments.
>
> Our method does not necessarily need to know a priori the meta-test class names but just the relations of novel categories. In such setting, any permutation of the 5 classes are still different task instances,  and thus the difficulty level of the problem is the same as that studied in previous few-shot learning works. WordNet (Miller, 1995) categorial taxonomy relations is an example we used for incorporating rich semantic knowledge in few-shot learning. It is similar to many existing work in zero-shot learning where the word embedding of novel classes are given a priori  for learning implicit relations of novel categories using pre-trained word embeddings (Socher et al., 2013;Frome et al., 2013), or explicit categorical relations such as the hierarchical structure of classes represented as knowledge graphs (Salakhutdinov et al., 2011; Deng et al., 2014). For example, (Wang et al., 2018) uses semantic embeddings of novel categories and the categorical relationships from WordNet knowledge graph to predict classifiers for zero-shot learning. (Kampffmeyer et al., 2018) follows this line and explores WordNet knowledge graphs of novel and base categories for zero-shot learning. In these methods, semantic features such as word embedding and knowledge graphs of novel categories are required as input.
>
> Compare to previous few-shot learning literature, our method indeed requires prior knowledge of novel category relations in addition to the few training samples of novel categories. However, this prior knowledge is general, universal, and can be easily obtained from rich textural semantic space (e.g., word embeddings trained from sufficiently large corpus, and WordNet taxonomy) a priori instead of hand-engineered. For example, the categorial relations in few-shot learning are borrowed from WordNet  taxonomy (Miller, 1995), and statistical computed from training data in medical abnormality and disease classification.
>
> The incorporation of prior categorical relations in few-shot learning highly aligns with human learning (Lake et al., 2015). For example, when teaching a baby what a bird is like, a teacher not only shows a few visual examples, but also provides verbal/textual descriptions of specific characteristics of the bird such that the baby can associate the novel category with categories that have already been learned. The learning process includes both visual understanding that is data-dependent, and contextual understanding which is universal and general.
>
> Intuitively, both universal and task-dependent categorical knowledge are beneficial for learning from limited examples. Graph Transformer demonstrates its superior capability of distilling information from prior universal knowledge, and effectively adapts itself for new training samples as specific tasks change. It also shows that the reconciliation of the universal knowledge-based and the task-dependent automatic learning are the key to success in empowering machine learning models in the few-shot learning task and the medical abnormality and disease classification task.
>
>
> References
> [1]  R. Socher, M. Ganjoo, C. D. Manning, and A. Y. Ng. Zero- Shot Learning Through Cross-Modal Transfer. In ICLR, 2013.
> [2] A. Frome, G. Corrado, J. Shlens, S. Bengio, J. Dean, and T. Mikolov. Devise: A deep visual-semantic embedding model. In NIPS, 2013.
> [3] R. Salakhutdinov, A. Torralba, and J. Tenenbaum. Learning to Share Visual Appearance for Multiclass Object Detection. In CVPR, 2011.
> [4] J. Deng, N. Ding, Y. Jia, A. Frome, K. Murphy, S. Bengio, Y. Li, H. Neven, and H. Adam. Large-Scale Object Classification Using Label Relation Graphs. In ECCV, 2014.
> [5] Wang, Xiaolong, Yufei Ye, and Abhinav Gupta. "Zero-shot Recognition via Semantic Embeddings and Knowledge Graphs." CVPR. 2018.
> [6] Kampffmeyer, Michael, et al. "Rethinking Knowledge Graph Propagation for Zero-Shot Learning." arXiv preprint arXiv:1805.11724 (2018).
> [7] Lake, Brenden M., Ruslan Salakhutdinov, and Joshua B. Tenenbaum. "Human-level concept learning through probabilistic program induction." Science 350.6266 (2015): 1332-1338.

---

> > ### Comment · AnonReviewer1 · 2018-11-30
> > **Could you point me to the SOTA clarifications in the manuscript?**
> >
> > Thanks for the clarifications in the comment, but I did not dispute that extra information is not available. I asked for clarifications to be added to the manuscript for your claims of being state-of-the-art on miniImageNet 5-way 1-shot which are simply not correct imho, since extra information was used. This extra information is not available to the other methods in your table, so the comparison is apples-to-oranges.
> >
> > I could not find any clarifications in the main text of the paper. Furthermore, moving the coefficient ablation to appendix further obfuscates just how important this extra information is for the result. This is worrying. I was expecting a honest discussion about how your method can achieve performance above SOTA in that domain using extra information, but is not strictly comparable and hence it is not the new SOTA. Could you please point me to sections where you cover this?

---

### Official Review · AnonReviewer3 · 2018-11-02
**Useful but straightforward idea**

**Rating:** 6
**Confidence:** 5

**Review:**

Summary
========
The  paper  adopts  the  self-attention  mechanism  in Transformer and in message-passing graph neural networks to derive  graph-to-graph mapping. Tested on few-shot learning, medical imaging classification and graph classification problems, the proposed methods show competitive performance.

Comment
========
Graph-to-graph mapping is an interesting setting and the paper presents an useful solution and interesting applications.  The paper is easy to read.

However, given the recent advancements in self-attention and message-passing graph modeling under various supervised settings (graph2vec, graph2set, graph2seq and graph2graph), the methodological novelty is somewhat limited. The idea of intra-graph and inter-graph message passing, for example, has been studied in:
Do et al. "Attentional Multilabel Learning over Graphs-A message passing approach." arXiv preprint arXiv:1804.00293 (2018).

Computationally, the current solution is not very scalable for large input and output graphs.

---

> ### Author Response · Authors · 2018-11-20
> **Response to Reviewer3**
>
> Thanks for the constructive comments.
>
> 1. Message passing including intra- and inter-graph is a common strategy in graph neural networks (Do et al.,2018; Gilmer et al., 2017; Schlichtkrull et al., 2018). However, the underlying design, formulation, and effectiveness of models can vary drastically. Our proposed GTR is novel in that it uses a multi-head global attention mechanism to enable global context encoding and parallelism. These technical merits mitigate the challenging vanished gradients problem in RNNs due to sequential processing, and enables long-term relationship modeling and fast training which RNNs and most GNNs with only local context encoding suffer. Besides, the second last paragraph in section 1. Introduction details GTR’s technical merits.
>
> We have added more discussions of relevant approaches in Appendices A, including Graph Attentional Multi-Label Learning (GAML) (Do et al.,2018), and other  previous work or parallel ICLR submissions on graph transformation such as graph2vec (Narayanan et al., 2016), graph2graph (https://openreview.net/forum?id=r1xYr3C5t7 ), and graph2seq (https://openreview.net/forum?id=SkeXehR9t7; https://openreview.net/forum?id=Ske7ToC5Km).
>
> 2. Graph Transformer is able to process large graph inputs as the sparsity among edges can be promoted. For example,  in the few-shot learning experiments, we propose to trim edges by a threshold, resulting in a sparse graph representation. The sparser the graphs are, the easier for GTR to extract features from the most related nodes, and the faster the information flows among strongly correlated nodes. The user-defined sparsity degree controls the scalability of GTR to large graphs, and the intensity of modeling long-term relationship between nodes, which is lacked in many GNNs (Kipf & Welling, 2017;Defferrard et al., 2016). In addition, different sparsity levels in different GTR layers can also be implemented, so as to gradually distill significant high-level semantics from large low-level input graphs.
>
> In our experiment of medical abnormality and disease classification on the CX-CHR dataset, the abnormality graph contains 155 nodes and the computation by our GTR model is fast. In contrast, in the experiments conducted in GAML (Do et al.,2018), the average numbers of nodes are only 27.68 and 25.31 for the 9cancers and 50proteins datasets, respectively, which are much smaller than ours.
>
> References
> [1] Dehghani, Mostafa, Stephan Gouws, Oriol Vinyals, Jakob Uszkoreit, and Łukasz Kaiser. "Universal transformers." arXiv preprint arXiv:1807.03819 (2018).

---

### Official Review · AnonReviewer2 · 2018-11-05
**This paper proposes an intereting method for graph dataset. However,  some points need to be verified.**

**Rating:** 6
**Confidence:** 5

**Review:**

This paper proposes a graph transformer method to learn features from the data with a graph structure. Actually it is the extension of Transformer network to the graph data. Although it is not very novel, yet it is interesting.  The experimental result has confirmed the author's claim.

I have some concerns as follows:
1. For the sequence input, this paper proposes to use the positional encoding as the standard Transformer network. However, for graphs, edges have encoded the relative position information. Is it necessary to incorporate this positional encoding? It's encouraged to conduct some experiments to verify it.

2. It is well known that graph neural networks usually have large memory overhead. How about this model? I found that the dataset used in this paper is not large. Can you conduct some experiments on large-scale datasets and show the memory overhead?

---

> ### Author Response · Authors · 2018-11-20
> **Response to Reviewer2**
>
> Thanks for the constructive comments.
>
> 1. We agree that for graph inputs such as knowledge graphs, positional encoding is not necessary since the input nodes are fixed and prior edges are given. We indeed do not use positional encoding in the experiments of knowledge graph transformation such as few-shot classification (section 4.1) and medical abnormality and disease classification (section 4.2), and graph representation learning (section 4.3).
>
> Positional encoding can be useful for sequence data (e.g., text) and visual data (e.g., image). For example, positional encoding can be used to encode object locations in an image. There is previous work (Liu et al., 2018) on incorporating positional cues such as coordinates as visual features to enhance representation learning and downstream tasks. We are happy to add ablation study on the effectiveness of positional encoding for visual features in our revised version.
>
> 2. In terms of memory overhead, GTR scales linearly with the graph size thanks to the proposed global attention mechanism. It does not require any additional memory compared to the standard Transformer (Vaswani et al., 2017) and recurrent neural networks. Please refer to Table 5 in the revised paper which summarizes the memory complexity for sequence and graph outputs.
>
> We compute memory usage of GTR and baseline model Gidaris & Komodakis (2018), and conduct ablation study on model size on 1-shot learning in Appendices E as well as theoretical analysis in Appendices B. Specifically, we study how memory usage and model performance change with parameter size by changing feature dimensions and hidden feature dimensions, and fixing all other hyper-parameters. The results and analysis is summarized in Table 7. Most importantly, GTR using 512 as feature dimension and hidden feature dimension (last row of Table 7) only consumes 0.0278G memory which is less than that used by Gidaris & Komodakis (2018) (0.0407G), and obtains much better results (60.6778±0.7095% v.s. 56.32±0.86%). This demonstrates that GTR is not only effective, but also memory efficient. Furthermore, all results are still larger than that of all baseline models, maintaining GTR’s state-of-the-art performance. Additionally, the memory usage for medical abnormality and disease classification (section 4.2) and graph classification experiment (section 4.3) is only 0.0476G and 0.0217G respectively.
>
> In terms of experiments on large datasets, the CX-CHR dataset used in the medical abnormality and disease classification is relatively large as it has 33,236 patient samples and 40,411 images in total where every patient can have multiple images. We are happy to conduct additional experiments on larger dataset.
>
> Last but not least, graph transformer holds several technical metrics that we missed to present in our initial version such as its time efficiency by requiring significantly less time to train than baseline methods, and only constant training and testing time for graph output. We add detailed explanation in Appendices C in our revised paper.
>
> References
> [1] Liu, Rosanne, Joel Lehman, Piero Molino, Felipe Petroski Such, Eric Frank, Alex Sergeev, and Jason Yosinski. "An intriguing failing of convolutional neural networks and the coordconv solution." arXiv preprint arXiv:1807.03247 (2018).

---

> > ### Comment · AnonReviewer2 · 2018-11-28
> > **Keep my score.**
> >
> > Thanks for author's feedback. Most of my concerns have been addressed.  I still have one more question. When you train GTR, do you use full gradient descent or mini-batch gradient descent method?

---

### Public Comment · (anonymous) · 2018-10-17
**Interesting work, questions about few-shot learning**

Thanks for the Graph Transformer work.

The idea of using both source attention and self-attention between different graphs is quite interesting and novel. Also, I like the natural layer stacking in Figure 1 (Right).

Specifically, I am interested in the few-shot learning experiments. In Table 1, the performance of GTR varies with different \lambda values.
My question is what is the accuracy of \lamba=0 and \lambda=1, since these two values indicate two special cases: only using predefined edge and only using learned edges.
Another question is about GTR for image input. In the paper, "each pixel is treated as graph nodes", how to get the class-level graph representation? Is it average pooling across pixels? And will this per-pixel nodes strategy increase the computation cost in graph operation?

---

> ### Author Response · Authors · 2018-10-22
> **Response**
>
> Thank you for your comment!
>
> The accuracies of 1-shot learning for \lambda=0.0 and 1.0 are 56.08+-0.79% and 61.56+-0.72% respectively. The highest 1-shot accuracy is 61.58+-0.71% achieved when \lambda=0.9. The accuracies of 5-shot learning for \lambda=0.0 and 1.0 are 72.12+-0.64% and 72.95+-0.64% respectively. The highest 5-shot accuracy is 73.21+-0.63% achieved when \lambda=0.7. It is worth noting that \lambda indicates the weight on prior edges. So \lambda=0.0 and 1.0 correspond to only using the learned edges, and only using the pre-defined edges respectively.
>
> Besides, the experiment shown in our paper was obtained only in the first round of our model, and used fixed learning rate 0.01 during episodic training. After allowing learning rate for episodic training to decrease 10 times whenever encountering validation performance plateau, we obtained the above improved results. The results demonstrate that: 1) GTR with prior category similarity improves few-shot learning performance over its direct baseline framework Gidaris & Komodakis (2018) on both 1-shot and 5-shot learning, and achieves the state-of-the-art performance on 1-shot learning; 2) 1-shot learning relies more on the prior knowledge of similarity between base and novel categories than 5-shot learning; 3) GTR(\lambda=0.0) is slightly lower than that achieved by Gidaris & Komodakis (2018), however not statistically significant, indicating that the attention mechanisms in both models have similar effectiveness in this task. We will add more ablation study in our revised version.
>
> For the question on GTR for image input, images are usually first fed to a deep network for features extraction, and the extracted features such as the output of the last convolutional layer of a deep network are then used as input of GTR. Thus, the visual input to GTR generally has small size such as 5*5*128 and 16*16*256 (thus, graph node size is 25 and 256). In graph classification task, we adopted naive average pooling for aggregating node features for class-level graph classification. However, other techniques such as cluster-based pooling can be incorporated.

---

### Comment · AnonReviewer1 · 2018-10-29
**How comparable are the few-shot learning results with other approaches?**

Thanks for an interesting paper with diverse experiments!

I wonder how comparable the few shot-learning results are to related work, since this paper claims state-of-the-art performance, but seems to use extra information about the meta-test classes, which would make results not directly comparable. It is my understanding that wordnet information is used to an increasing extent as \lambda gets closer to 1, and that a \lambda value of 0 is the only truly comparable result. Furthermore, this approach seems to assume that meta-test class names are known, which is not commonly assumed in other approaches. Indeed, most approaches would have no choice but to consider any permutation of the 5 classes as different task instances, which is (arguably) a harder problem. Could you please clarify these details?

---

### Public Comment · (anonymous) · 2018-11-19
**Missing related work in experiments.**

While the proposed solution was compared to a few algorithms, some recent state-of-the-art algorithms were omitted in the experiments sections, having a misleading impression on the performance of the author's algorithm. At least the following papers should be included and argued the differences with the author's approach.

[1] Ivanov et.al, Anonymous Walk Embeddings, ICML 2018
[2] Verma et.al, Hunt For The Unique, Stable, Sparse And Fast Feature Learning On Graphs, NIPS 2017

---

> ### Author Response · Authors · 2018-11-20
> **Response to "Missing related work in experiments"**
>
> Thanks for the comment. The two papers mentioned have conducted experiments on PROTEINS and D&D dataset for graph classification, and both of them obtained lower performance than our graph transformer (GTR). Specifically, [1] obtained 71.51 ± 4.02 on D&D, [2] obtained 77.10 on D&D and 73.42 on PROTEINS. Graph transformer obtained 79.15 on D&D and 75.70 on PROTEINS, surpassing both compared models by large margins.
>
> Besides, [1] uses random walk for graph representation learning which is limited to local context encoding, and only uses predefined edge weights, while GTR enables global context encoding via global attention mechanism and incorporation of predefined edge weights and learnable edge weights. [2] uses graph spectral distances for learning graph features which requires complex computation such as eigendecomposion while our method does not require such cost. Additionally, both [1] and [2] use SVM as algorithm for graph classification, while our method does not use any additional classifier.
>
> Graph neural modeling is an active research field with emergingly many new methods. We have compared with 10 previous methods including 4 kernel-based methods and 6 GNN-based methods in our graph classification experiments. We are happy to include the above results, and make comparison with more previous work to demonstrate the performance of our approach.
>
> [1] Ivanov et.al, Anonymous Walk Embeddings, ICML 2018
> [2] Verma et.al, Hunt For The Unique, Stable, Sparse And Fast Feature Learning On Graphs, NIPS 2017

---

### Author Response · Authors · 2018-11-26
**Summary of author response**

We thank all reviewers for their constructive comments. We have updated the paper with Appendix A-E to address reviewers' concerns and further demonstrate Graph Transformer’s (GTR) performance. Here is a summary of these updates:

(1) R1 expressed concern on comparison of GTR with previous methods which do not require the relations of meta-test and base classes a priori. We argue that the relation of categories GTR requires are universal and general (e.g., borrowed from WordNet taxonomy (Miller, 1995)), and the usage of which inherently follows human learning where reconciliation of data-dependent visual recognition and universal knowledge happens. Besides, WordNet taxonomy knowledge graph (Miller, 1995) is adopted in many related zero-shot learning papers (Wang et al., 2018; Kampffmeyer et al., 2018; Salakhutdinov et al., 2011) as prior meta-test and base classes relations. Our experiment proves that this reconciliation leads to stronger performance in the challenging few-shot setting, and the learning of complex causal relations among medical abnormalities and diseases which is essential for explanatory diagnosis.

(2) To address R2’s concern on memory overhead of GTR, we provided theoretical analysis of memory usage in Appendix B, as well as experimental results on the relation between memory usage, model performance and model size in Appendix E. Our experiment shows that GTR is capable of achieving state-of-the-art performance on 1-shot classification using less memory and parameters than its direct baseline model Gidaris & Komodakis (2018).

(3)To address R3’s concern on computational efficiency of GTR, we provided theoretical analysis on time complexity in Appendix C. Our analysis shows that with sequences and graphs as output, GTR costs constant training time, and linear to the output size and constant testing time respectively. Thanks to GTR’s global attention mechanism, it has higher computational efficiency than recurrent neural network, and is capable of scaling to large graphs. The scaling capability is further enhanced by our proposed trimming scheme which promotes sparsity of edges.

(4) We additional conducted ablation study on weight of prior edges (\lambda) in few-shot learning, and provided results in Appendix D. Our experiment shows that prior relations of meta-test classes are beneficial, and the combination of which and automatic learning leads to the best performance.

(5) Besides the 10 baselines we have already compared with in graph classification task, we additionally compared with AWE (Ivanov & Burnaev, 2018) and FGSD (Verma & Zhang, 2017) in section 4.3 Table 4, and showed improved performance. We have also added more discussion of recent related work in Appendix A, including AWE (Ivanov & Burnaev, 2018), FGSD (Verma & Zhang, 2017), GAML (Do et al.,2018), graph2vec (Narayanan et al., 2016), graph2graph (https://openreview.net/forum?id=r1xYr3C5t7), and graph2seq (https://openreview.net/forum?id=SkeXehR9t7; https://openreview.net/forum?id=Ske7ToC5Km), and highlighted the difference and advantages of the proposed approach.

---

### Meta-Review · Area_Chair1 · 2018-12-15
**Reviewers agree that work is interesting, but reviews are borderline**

**Confidence:** 4
**Recommendation:** Reject

**Metareview:**

The reviewers all agree that the work is interesting, but none have stood out and championed the paper as exceptional. The reviewers note that the paper is well-written, contributes a methodological innovation, and provides compelling experiments. However, given the reviewers' positive but unenthusiastic scores, and after discussion with PCs, this paper does not meet the bar for acceptance into ICLR.